# Transcriptome Profiling of Primary Skin Fibroblasts Reveal Distinct Molecular Features Between *PLOD1*- and *FKBP14*-Kyphoscoliotic Ehlers–Danlos Syndrome

**DOI:** 10.3390/genes10070517

**Published:** 2019-07-08

**Authors:** Pei Jin Lim, Uschi Lindert, Lennart Opitz, Ingrid Hausser, Marianne Rohrbach, Cecilia Giunta

**Affiliations:** 1Connective Tissue Unit, Division of Metabolism and Children’s Research Centre, University Children’s Hospital, 8032 Zürich, Switzerland; 2Functional Genomics Center Zurich, University of Zurich/ETH Zurich, Winterthurerstrasse 190, 8057 Zürich, Switzerland; 3Institute of Pathology, Heidelberg University Hospital, 69120 Heidelberg, Germany

**Keywords:** kyphoscoliotic Ehlers–Danlos Syndrome, EDS type VI, transcriptomics, connective tissue, extracellular matrix, *PLOD1*, *FKBP14*

## Abstract

Kyphoscoliotic Ehlers–Danlos Syndrome (kEDS) is a rare genetic heterogeneous disease clinically characterized by congenital muscle hypotonia, kyphoscoliosis, and joint hypermobility. kEDS is caused by biallelic pathogenic variants in either *PLOD1* or *FKBP14*. *PLOD1* encodes the lysyl hydroxylase 1 enzyme responsible for hydroxylating lysyl residues in the collagen helix, which undergo glycosylation and form crosslinks in the extracellular matrix thus contributing to collagen fibril strength. *FKBP14* encodes a peptidyl-prolyl cis–trans isomerase that catalyzes collagen folding and acts as a chaperone for types III, VI, and X collagen. Despite genetic heterogeneity, affected patients with mutations in either *PLOD1* or *FKBP14* are clinically indistinguishable. We aim to better understand the pathomechanism of kEDS to characterize distinguishing and overlapping molecular features underlying *PLOD1*-kEDS and *FKBP14*-kEDS, and to identify novel molecular targets that may expand treatment strategies. Transcriptome profiling by RNA sequencing of patient-derived skin fibroblasts revealed differential expression of genes encoding extracellular matrix components that are unique between *PLOD1*-kEDS and *FKBP14*-kEDS. Furthermore, we identified genes involved in inner ear development, vascular remodeling, endoplasmic reticulum (ER) stress, and protein trafficking that were differentially expressed in patient fibroblasts compared to controls. Overall, our study presents the first transcriptomics data in kEDS revealing distinct molecular features between *PLOD1*-kEDS and *FKBP14*-kEDS, and serves as a tool to better understand the disease.

## 1. Introduction

According to the 2017 revised Nosology of the Ehlers–Danlos syndrome (EDS) [1], kyphoscoliotic EDS (kEDS, OMIM 225400 and 614557) groups two rare autosomal recessive disorders which are clinically indistinguishable, but genetically distinct as they are caused by pathologic biallelic variants in either procollagen-lysine,2-oxoglutarate 5-dioxygenase 1 (*PLOD1*) or FK506-binding protein 14 (*FKBP14*). *PLOD1* encodes the lysyl hydroxylase 1 (LH1) enzyme, which hydroxylates lysyl residues of Xaa-Lys-Gly tripeptide motif in collagens. Subsequently, the hydroxylated lysyl residues undergo glycosylation with the attachment of galactose or glucosyl-galactose units and after collagen secretion into the extracellular matrix (ECM) they form inter- and intra-molecular crosslinks known to contribute to collagen fibril strength and thus to tissue stability [2]. The lack or loss of function of LH1 leads to underhydroxylation and underglycosylation of lysyl residues in the helical domain of collagen, thereby impairing collagen cross-linking and consequentially causing mechanical instability of the affected connective tissues. Patients deficient in LH1 have an increased ratio of urinary lysyl-pyridinoline to hydroxylysyl-pyridinoline (LP/HP) due to underhydroxylation of collagen lysyl residues. Only recently, pathogenic variants in *FKBP14* have been described in a group of patients with a clinical diagnosis of kEDS but a normal LP/HP ratio [3,4]. *FKBP14* encodes the endoplasmic reticulum (ER)-resident FKBP22 protein, a 22kDa member of the family of FK506-binding peptidyl-prolyl cis–trans isomerases. FKBP22 catalyzes the cis–trans isomerization of prolyl peptide bonds, which is the rate limiting step in procollagen protein folding due to its abundance in proline residues, and has been shown to catalyze the folding of collagen type III [5]. FKBP22 also functions as a chaperone for collagen types III, VI and X where it is thought to prevent premature interactions between collagen during its assembly in the ER [5].

Despite harboring mutations in two genes with different functions, *PLOD1*-kEDS and *FKBP14*-kEDS patients share a common clinical phenotype with major criteria consisting of congenital muscle hypotonia, congenital or early onset kyphoscoliosis, and generalized joint hypermobility; shared minor criteria include skin hyperextensibility, rupture, or aneurysm of medium-sized arteries, easy bruising of skin, and osteopenia or osteoporosis. Gene-specific minor criteria for *PLOD1*-kEDS include skin fragility, while that of *FKBP14*-kEDS include congenital hearing impairment [1]. Pronounced inter- and intrafamilial variability of the clinical presentation have been described in both genetic forms of kEDS [3,4]. In the absence of genetic information, collagen type VI-related myopathies which present with severe neonatal hypotonia, delayed motor development, kyphoscoliosis and joint hypermobility [6,7,8] represent the major differential diagnosis to kEDS. 

Currently, there is no pharmacological treatment for kEDS and the treatment of manifestations aims to alleviate symptoms and focuses on the musculoskeletal and hearing systems [3,4,9]: physical therapy for muscular hypotonia, bracing of unstable joints, corrective surgeries for kyphoscoliosis and prescription of hearing aids for hearing impairment. 

The aim of our study is to better understand the pathomechanism of kEDS and to identify overlapping and distinguishing molecular signatures of *PLOD1*-kEDS and *FKBP14*-kEDS. The application of an untargeted general omics approach may serve as a valuable tool to identify novel proteins or pathways involved in the pathogenesis of kEDS that can be pharmacologically targeted to improve the disease symptoms. As a first step towards this goal, by performing RNA sequencing on an in vitro cell culture model of kEDS we obtained and compared the transcriptome profiles of normal control, *PLOD1*-kEDS and *FKBP14*-kEDS patients. Furthermore, the transcriptome profiles of kEDS patients were compared to published transcriptomics data of patients with collagen VI-associated myopathies [10]. 

## 2. Materials and Methods 

### 2.1. Subjects and Cell Culture 

This study was conducted according to the Declaration of Helsinki for Human Rights and approved by Swiss Ethics (KEK-ZH-Nr. 2019-00811) in the presence of a signed informed consent of the patients or their parents.

As part of the diagnostic workup of kEDS, punch biopsies of the skin for electron microscopy investigations and establishment of fibroblast cultures were previously obtained. The biological material was stored in the Biobank of the Division of Metabolism at the Children’s Hospital Zurich. For this study, fibroblasts of three *FKBP14*-kEDS patients, three *PLOD1*-kEDS patients and four healthy controls were used (Appendix A). Pathological mutations and clinical findings of the patients are recorded in Table 1. Cells were cultured at 37 °C and 5% CO_2_ in Dulbecco’s Modified Eagle’s Medium (Gibco, 31966-021) supplemented with 10% fetal bovine serum, 100 U/ml penicillin, 100 mg/ml streptomycin, and 0.25 mg/ml Amphotericin B.

### 2.2. Gene Expression Profiling

Cells were passaged into T75 flasks, fed fresh medium 24 hours after passaging, and RNA was isolated 24 hours later for transcriptome profiling. RNA was harvested using miRNeasy Mini Kit (QIAGEN, 217004) according to the manufacturer’s instructions. RNA quality control was performed on an Agilent 2100 Bioanalyzer with RNA integrity number (RIN) values between 9.6 and 10.0. Poly-A purified libraries were prepared using the TruSeq mRNA sample preparation kit (Illumina, 20020595). RNA sequencing was performed on an Illumina HiSeq 4000 instrument at the Functional Genomics Center Zurich. The raw reads were cleaned by removing adapter sequences, trimming low quality ends, and filtering reads with low quality (phred quality <20). Sequence alignment of the resulting high-quality reads to the human genome (build GRCh38.p10) and quantification of transcript expression was carried out using RNA-Seq by Expectation Maximization (RSEM, version 1.3.0) with Ensembl gene models of release 89. A count-based negative binomial model implemented in the software package edgeR (R-version 3.5.1, edgeR 3.24.2) was applied to detect differentially expressed genes (DEGs). DEGs were defined as genes with *p*-value <0.05 and log_2_(fold change) >0.5 or <−0.5. To identify enriched biological processes and cellular components, over-representation enrichment analysis (ORA) using a list of DEGs generated with a more stringent cutoff of *p*-value <0.01 and gene set enrichment analysis (GSEA) using ranked gene lists were performed with the online toolkit WebGestalt (2017 version) [13,14]. The RNA sequencing data, including raw sequence files for each subject, is available on the European Nucleotide Archive (ENA) and the Gene Expression Omnibus (GEO) database under the accession number PRJEB31335.

### 2.3. Quantitative RT-PCR for Validation

Candidate genes were selected from the lists of DEGs for validation by quantitative RT-PCR in four independent replicates. RNA was harvested using RNeasy Mini Kit (QIAGEN, 74104) according to manufacturer’s instructions, and reverse transcribed to cDNA using the High-Capacity RNA-to-cDNA Kit (Applied Biosysyems, 4387406). cDNA was diluted in RNase-free water to 3 ng/µl for quantitative RT-PCR using Taqman assays (Table 2) on a 7900HT Fast Real-Time PCR System machine (Applied Biosystems). Fold change in gene expression was calculated by the 2^–∆∆Ct^ method with glyceraldehyde-3-phosphate dehydrogenase (*GAPDH*) as an endogenous control.

## 3. Results

### 3.1. Transcriptome Profiling and Differential Expression Analysis

Transcriptome profiling by RNA sequencing identified 298 DEGs in *PLOD1*-kEDS patient-derived fibroblasts compared to controls, of which 139 genes were up-regulated and 159 genes were down-regulated in the patients’ cells, as represented in a volcano plot (Figure 1A, Appendix A). 488 DEGs were identified in *FKBP14*-kEDS patient-derived fibroblasts compared to controls, of which 309 genes were up-regulated and 179 genes were down-regulated (Figure 1B, Appendix A). The low numbers of DEGs observed in our datasets may be contributed by heterogeneity in multiple factors such as ethnicity, age of biopsy, and sex of patients. Nevertheless, several genes encoding proteins related to ECM composition, protein trafficking, vasculature development, and inner ear development were identified in the lists of DEGs, which can potentially contribute to the pathogenesis as discussed later in this paper.

To further explore the similarities and differences between *PLOD1*-kEDS and *FKBP14*-kEDS patient-derived fibroblasts, we compared DEGs with *p* < 0.05 and log_2_ (fold change) >0.5 or <−0.5 in both patient groups, and found 58 overlapping genes (Figure 1C). Of these 58 genes, 37 were up-regulated (Figure 1D) and 18 were down-regulated (Figure 1E) in both *PLOD1*-kEDS and *FKBP14*-kEDS compared to controls. Furthermore, from the more stringent list of DEGs with *p* < 0.01, 11 genes overlapped between *PLOD1*-kEDS and *FKBP14*-kEDS fibroblasts (Figure 1F). Of these 11 DEGs, 9 genes were up-regulated (Figure 1G) in both groups, 1 gene was down-regulated (Figure 1H) in both groups, and 1 gene was up-regulated in *PLOD1*-kEDS but down-regulated in *FKBP14*-kEDS fibroblasts (further discussed in Figure 2A).

### 3.2. Gene Ontology and Pathway Analyses

Due to limitations in obtaining patient skin fibroblasts in a rare disease with newly described gene mutations, the sample size in our study was inevitably inadequate to generate a list of DEGs adjusted for false discovery rate (FDR). Instead, using the WebGestalt toolkit, we analyzed the RNA sequencing datasets with two approaches: (i) by setting a more stringent *p*-value cutoff whereby DEGs with *p* < 0.01 in *PLOD1*-kEDS and *FKBP14*-kEDS patient-derived fibroblasts were used to identify over-represented biological processes and cellular components by ORA and (ii) by performing GSEA in which a list of all detected genes ranked by their average fold change is used to calculate enrichment scores of Kyoto Encyclopedia of Genes and Genomes (KEGG) pathways. Gene ontology terms with an FDR <10% were considered significantly enriched in both approaches.

In the ORA using DEGs with *p* < 0.01, the significantly enriched biological processes and cellular components are summarized in Table 3; DEGs that contribute to these enriched Gene Ontology (GO) terms are listed in Appendix A. Notably, genes encoding ECM proteins were enriched in the DEGs in both *PLOD1*-kEDS and *FKBP14*-kEDS patient-derived fibroblasts; each patient group was compared to control fibroblasts. However, comparison of DEGs that contributed to over-representation of ECM components demonstrated only one overlapping gene, *EFEMP1*, which was up-regulated in *PLOD1*-kEDS and down-regulated in *FKBP14*-kEDS fibroblasts (Figure 2A). The expression levels of each DEG encoding ECM components in individual control and patient-derived fibroblasts are depicted in a heatmap (Figure 2B). To further investigate the differential expression in ECM-related genes between *PLOD1*-kEDS and *FKBP14*-kEDS, we performed ORA of the DEGs between *PLOD1*-kEDS and *FKBP14*-kEDS fibroblasts with p-value <0.01 and log_2_ (fold change) >0.5 or <−0.5 (Appendix A) using WebGestalt and observed an enrichment of ECM genes (GO:0005578) with an enrichment ratio of 7.64 and FDR of 6.41 × 10^−5^. The expression of these ECM genes that are significantly different between *PLOD1*-kEDS and *FKBP14*-kEDS are summarized in a heatmap (Figure 2C). Hence, the RNA sequencing datasets revealed distinct ECM signatures between *PLOD1*-kEDS and *FKBP14*-kEDS fibroblasts at the transcriptome level.

GSEA of the ranked gene lists from *PLOD1*-kEDS fibroblasts did not identify significantly enriched KEGG pathways. In contrast, KEGG pathways involved in carbohydrate metabolism, namely fructose and mannose metabolism (Figure 3A) as well as glycolysis and gluconeogenesis (Figure 3B), were positively enriched in *FKBP14*-kEDS fibroblasts. Furthermore, genes involved in cell cycle (Figure 3C) and DNA replication (Figure 3D) were over-represented by down-regulated genes in *FKBP14*-kEDS fibroblasts, suggesting alterations in cell proliferation.

The findings from the ORA and GSEA analyses collectively suggests different molecular signatures between *PLOD1*-kEDS and *FKBP14*-kEDS.

### 3.3. Comparison of Transcriptome Profiles between kEDS and Collagen VI-Related Muscular Dystrophies

Due to the overlaps in clinical presentation between collagen VI-related muscular dystrophies (COL6-RD) and kEDS patients, we compared the transcriptome profiles of patient-derived fibroblasts from *PLOD1*-kEDS and *FKBP14*-kEDS (DEGs with *p* < 0.01) with that of COL6-RD patients available through GEO Series accession number GSE103270 [10]. Only one gene, *OLFM2*, was significantly up-regulated in all three patient groups; *EFEMP1* was significantly up-regulated in *PLOD1*-kEDS and COL6-RD but down-regulated in *FKBP14*-kEDS (Figure 4A). While the ECM component, which is biologically relevant to disorders of the connective and muscle tissues, was over-represented in the DEGs from all three groups (Table 3 and [10]), comparison of DEGs encoding ECM components also revealed only a few overlapping genes among the three patient groups (Figure 4B). Furthermore, the over-representation of muscle system process in *FKBP14*-kEDS fibroblasts (GO:0003012, Table 3) prompted us to compare the genes contributing to this GO term in *FKBP14*-kEDS to the DEGs in COL6-RD fibroblasts. Of the nine genes contributing to the enrichment of muscle system process, only two were significantly differentially regulated in COL6-RD fibroblasts carrying dominant negative (DN) collagen VI mutations; six of these nine genes (three at *p* < 0.01, three at *p* < 0.05) were also differentially regulated in *PLOD1*-kEDS (Figure 4C). These suggest that each disease group have a unique transcriptome profile. 

### 3.4. DEGs with Biological Functions that May Contribute to the Pathogenesis of kEDS

From our transcriptomics analysis, we also identified several DEGs with biological functions that may contribute to the pathogenesis of kEDS. We validated the differential expression of these candidate genes by quantitative RT-PCR (Figure 5). How the encoded products of these genes could contribute to the disease pathology is further discussed under Section 4.1 to Section 4.6.

## 4. Discussion

kEDS represents a paradigm heterogeneous connective tissue disorder in that genetic defects in two different proteins, LH1 and FKBP22, involved in the biosynthesis of collagens lead to a similar clinical phenotype. As an explanation thereof, we hypothesize that *PLOD1* and *FKBP14* mutations might lead to similar alterations of the ECM of connective tissue, either by affecting the same molecular pathways, or by converging the different altered pathways to the same target molecules. To answer this question, we have applied an untargeted omics approach to the identification of common and distinct molecular pathways altered in *FKBP14*-kEDS and *PLOD1*-kEDS. 

Our studies were performed in patient-derived primary skin fibroblasts, which are clinically relevant to connective tissue disorders since they are the major cell type producing ECM proteins, and are relatively easy to obtain via skin biopsies. We do, however, acknowledge that clinical characteristics of kEDS patients extend beyond abnormalities in the skin and that the functions of some DEGs may appear more relevant in other ECM-producing cells, including chondrocytes, osteoblasts, and muscle fibroblasts. Nevertheless, the transcriptomics datasets will help to identify candidate DEGs for functional characterization when the other cell types or animal models eventually become available. 

Here, we present the outcome of our investigations by transcriptomics of cultured dermal fibroblasts from individuals with *FKBP14*-kEDS and *PLOD1*-kEDS which led to the identification of over-represented biological processes and cellular components. By GSEA, we observed positive enrichment of fructose and mannose metabolism, as well as glycolysis and gluconeogenesis in *FKBP14*-kEDS fibroblasts. Additionally, cell cycle and DNA replication were negatively enriched in *FKBP14*-kEDS fibroblasts, suggesting that cells deficient in FKBP22 may undergo a metabolic switch and alteration in cell proliferation. In vitro studies using mammary epithelial cells have shown that changes in the density of the ECM lead to alterations in glucose metabolism [15]. In another study, up-regulation of glycolysis was observed in healthy skin from the footpad of mice with a dense ECM compared to the abdominal skin with a thinner ECM; glycolysis was also up-regulated in fibrotic skin compared to healthy skin [16]. Reciprocally, perturbations in metabolism can cause alterations in the ECM. In particular, gene and protein expression of ECM components are down-regulated upon suppression of glycolysis [16]. As such, it will be interesting to investigate whether the abnormal deposition of ECM proteins by FKBP22-deficient cells as previously shown [4] cause metabolic changes which can eventually alter cellular bioenergetics and proliferation, or whether alterations in metabolism affects the formation of the ECM by the fibroblasts. 

From the transcriptome profiling analysis, we also observed differential expression of several genes encoding products with biological functions that may contribute to the pathogenesis of kEDS (Figure 5). In the following sections, we discuss how the encoded products of these DEGs could contribute to the disease pathology.

### 4.1. ECM Components

Gene ontology analysis showed significant enrichment of the ECM by DEGs in both *PLOD1*-kEDS and *FKBP14*-kEDS. We validated the up-regulation of *ELN* (encoding elastin), *POSTN* (encoding periostin) and *WNT4* (encoding Wnt family member 4) specifically in *PLOD1*-kEDS (Figure 5A).

Elastin, like collagens, is a structural component of the ECM that is rich in glycine and proline. However, elastin adopts random coil conformations rather than a triple helix structure that give rise to its stretchable properties.

Periostin is preferentially expressed in connective tissues under constant mechanical stress. It accelerates collagen cross-linking through promoting the proteolytic activation of lysyl oxidase (LOX) by bone morphogenetic protein-1 (BMP-1) and serves as a scaffold for collagen and BMP-1 [17]. Periostin also facilitates the formation of ECM meshwork by interacting with other ECM components, including fibronectin and tenascin-C [18]. The protective role of periostin is demonstrated by its induction upon skin injury to promote wound healing [19,20,21]. Overexpression of periostin has also been described in hypertrophic scar and keloid formation [22] and linked to the pathogenesis of skin fibrosis in systemic sclerosis [23,24]. In contrast, despite elevated *POSTN* expression in the skin fibroblasts (Table 1 and Figure 5A), *PLOD1*-kEDS patients present with poor wound healing and atrophic scarring. A plausible explanation for this observation is the under-hydroxylation of lysine residues in the collagen helical domain due to LH1 deficiency, which impairs collagen cross-linking and fibril assembly, and cannot be compensated for by the activation of LOX by periostin.

The physiological role of WNT4 has been described in the formation of synovial joints [25] and neuromuscular junctions [26]. Up-regulation of *Wnt4* expression during wound healing was demonstrated in a mouse model [27], which could facilitate wound closure by promoting cell migration [28]. 

Notably, the transcript levels of *ELN*, *POSTN* and *WNT4* can be up-regulated in response to transforming growth factor-beta1 (TGF-β1) stimulation [29,30,31]. Hence, it remains to be elucidated whether the elevated expression of *ELN*, *POSTN* and *WNT4* is a consequence of enhanced TGF-β signaling in *PLOD1*-kEDS patient fibroblasts, and if so, via which mechanism(s) TGF-β signaling is enhanced.

The down-regulation of *COL15A1* (encoding collagen type XV alpha 1 chain) in *FKBP14*-kEDS was confirmed by quantitative RT-PCR (Figure 5B). Type XV collagen is present widely in the basement membrane zones of cardiac and skeletal myocytes [32]. Studies performed in *Col15a1*-knockout mice demonstrated that deficiency in type XV collagen led to progressive skeletal muscle degeneration and variation in muscle fiber size starting from 13 weeks of age, and increased susceptibility to exercise-induced muscle injury [33]. However, the absence of these abnormal muscular phenotypes in newborn mice suggest that suppressed *COL15A1* expression unlikely explains for the presentation of congenital muscle hypotonia in *FKBP14*-kEDS patients. Nevertheless, rescuing *COL15A1* expression could be a potential therapeutic target to prevent worsening of the muscular phenotype with age in *FKBP14*-kEDS patients.

Interestingly, *EFEMP1* is down-regulated in *FKBP14*-kEDS patient-derived fibroblasts (Figure 5C). *EFEMP1* encodes the EGF-containing fibulin-like extracellular matrix protein, also known as fibulin-3. It is noteworthy that 47% of the *FKBP14*-kEDS patients described [3,4] developed inguinal or umbilical hernias, a phenotype that is also observed in *Efemp1*-knockout mice. Moreover, *Efemp1*-knockout mice have less elastic fibers in the dermis, which also appear more fragmented than elastic fibers in wildtype mice [34]. Therefore, we examined electron micrographs of skin biopsies taken from a control, a *PLOD1*-kEDS patient and two *FKBP14*-kEDS patients and observed fragmentation of elastic fibers in *FKBP14*-kEDS patients but not in the *PLOD1*-kEDS patient (Figure 6). Furthermore, a reduced incorporation of elastin into the network of elastic fibers in *FKBP14*-kEDS patients was observed. These observations, in concordance with the observations by McLaughlin and co-workers in *Efemp1*-knockout mice, suggest that fibulin-3 contributes to the maintenance of ECM integrity and particularly in the formation of elastic fibers. We are currently investigating whether there is indeed a reduction in fibulin-3 protein secretion by *FKBP14*-kEDS fibroblasts in vitro, and whether addition of recombinant fibulin-3 protein can overcome the deficiency and facilitate the formation of elastic fibers in an in vitro model.

Contrary to *EFEMP1* down-regulation in *FKBP14*-kEDS, *EFEMP1* was up-regulated in *PLOD1*-kEDS. The overexpression of *EFEMP1* in a chondrogenic cell line in vitro led to suppressed chondrocyte differentiation and dampened expression of aggrecan and types II and X collagen [35]. Thus, whether elevated expression of *EFEMP1* contributes to the pathogenesis of *PLOD1*-kEDS via altering cartilage development remains to be explored, despite the absence of reports on cartilage growth-plate pathologies in both human patients and *Plod1*-knockout mice. 

### 4.2. Inner Ear Development

Sensorineural hearing loss has been reported in 73% of *FKBP14*-kEDS patients [3,4], but not in *PLOD1*-kEDS patients. The transcriptome profiles showed an elevated expression of *ALDH1A3* encoding aldehyde dehydrogenase 1 family member A3 enzyme in *FKBP14*-kEDS patient-derived fibroblasts only, which was confirmed by quantitative RT-PCR (Figure 5D). During normal inner ear canal formation, *Aldh1a3* is repressed by the chromatin remodeler Chd7. Deficiency in Chd7 causes abnormally high levels of *Aldh1a3* expression and inner ear malformation; inner ear development is rescued by the loss of *Aldh1a3* expression in mice deficient in Chd7 [36]. Hence, we postulate a molecular link between enhanced *ALDH1A3* expression and hearing loss in *FKBP14*-kEDS patients that warrants further investigation. 

### 4.3. Vasculature Integrity

We sought to identify DEGs in *PLOD1*-kEDS and *FKBP14*-kEDS that are involved in modulating the vasculature, since rupture or aneurysm of medium-sized arteries have been described in both *PLOD1*-kEDS [12,37] and *FKBP14*-kEDS [3,38,39] and serves as a minor criteria in the diagnosis of kEDS [1]. We observed an up-regulation of *OLFM2* expression (encoding olfactomedin 2) in both *PLOD1*-kEDS and *FKBP14*-kEDS patient-derived fibroblasts (Figure 5E). *OLFM2* is also significantly up-regulated in COL6-RD patient-derived fibroblast [10], although vascular complications involving medium-sized arteries have not been reported in COL6-RD to our knowledge; instead, fenestration and narrow lumens in capillaries have been described in some cases of Ulrich congenital muscular dystrophy [40]. *Olfm2* is up-regulated in balloon-injured arteries and involved in smooth muscle cell phenotypic modulation and vascular remodeling. *Olfm2*-deficient mice are more protected against injury-induced suppression of smooth muscle cell markers and neointimal hyperplasia [41], suggesting that enhanced expression of *OLFM2* may correlate with poorer vascular phenotypes. Notably, *OLFM2* expression was not significantly altered in fibroblasts of vascular EDS patients [42]. Hence, whether elevated *OLFM2* expression contributes to the causation and/or severity of vascular complications in kEDS remains an interesting topic to be explored. The up-regulation of *OLFM2* in *PLOD1*-kEDS, *FKBP14*-kEDS, and COL6-RD also warrants further investigation into whether olfactomedin 2 plays a physiological role in skeletal muscles, which has not yet been described.

In addition, we noted that *TM4SF1* expression was down-regulated in *FKBP14*-kEDS and up-regulated in *PLOD1*-kEDS (Figure 5E). *TM4SF1* encodes the transmembrane-4-L-six-family-1 protein, also known as L6 cell surface antigen, and is highly expressed in the vascular endothelium. In a previous study, knockdown of *TM4SF1* in human umbilical vein endothelial cells (HUVEC cell line) resulted in a senescence phenotype and poor migration in a wound healing assay in vitro. Moreover, the authors showed that the angiogenesis inducer Vascular Endothelial Growth Factor A (VEGF-A) enhanced *Tm4sf1* expression in vivo in mice, while knockdown of *Tm4sf1* blunted the angiogenic effect of VEGF-A, thus demonstrating the importance of TM4SF1 in vessel maturation [43]. As such, it would be interesting to determine if the dysregulation of *TM4SF1* expression contributes to vascular fragility in kEDS. 

### 4.4. Unfolded Protein Response (UPR)

Another striking observation described in the first cohort of *FKBP14*-kEDS patients is the enlargement of ER in skin biopsies [4]. This dilatation of ER is thought to arise from the accumulation of misfolded proteins, in particular of collagens since the rate limiting step in collagen protein folding involves cis–trans isomerization of the prolyl peptide bonds which is catalyzed by FKBP22. ER dilatation coupled with the induction of UPR involving binding immunoglobulin protein (BiP) activation has been demonstrated in several models of connective tissue disorders, including (i) the Osteogenesis Imperfecta (OI) *Aga2* mouse model harboring *Col1a1* mutations which ablates the conserved C-terminal cysteine C244 residue and introduces additional amino acids into the C-terminal propeptide [44], (ii) in skin fibroblasts isolated from OI patients carrying *COL1A1* mutations in the C-terminal propeptide domain [45], and (iii) in engineered cells with *COL10A1* mutations modelling Schmid metaphyseal chondrodysplasia [46]. However, the transcriptome profiles and gene ontology analyses revealed that genes associated with classical ER associated degradation and UPR, including *CANX*, *CALR*, *HSPA5*, *ATF4*, and *DDIT3*, were not differentially expressed in *FKBP14*-kEDS patient-derived fibroblasts; the same trend was observed in *PLOD1*-kEDS patient-derived fibroblasts (Appendix A). The absence of classical UPR activation was also previously observed in other connective tissue disorders, such as in OI patients with mutations in *COL1A1* disrupting triple helix formation [45] and in vascular EDS patients carrying *COL3A1* mutations [42]. 

Nevertheless, one of the most up-regulated genes in *FKBP14*-kEDS patient-derived fibroblasts was *SCAMP5* encoding the secretory carrier-associated membrane protein 5, which was also induced in *PLOD1*-kEDS patient-derived fibroblasts albeit at a smaller magnitude (Figure 5F). *SCAMP5* expression can be rapidly induced by autophagic stimulation under the control of the master autophagy transcriptional regulator transcription factor EB (TFEB), and its elevated expression was previously described in the striatum of Huntington disease patients [47,48]. Interestingly, SCAMP5 enhances the aggregation of mutant huntingtin by impairing endocytosis [47], yet promotes Golgi fragmentation and unconventional secretion of α-synuclein via exosomes [48]. This warrants for future exploration into the role of SCAMP5 in kEDS patient-derived fibroblasts in relation to ER stress, autophagy and whether it promotes the retention or non-classical secretion of misfolded collagen proteins.

### 4.5. Bone Remodeling

Osteopenia or osteoporosis is a minor criteria in the diagnosis of kEDS [1], and occurrence of fractures were described in 13% of *FKBP14*-kEDS [3,4]. We observed elevated expression of *FGF11*, which encodes the intracellular fibroblast growth factor 11, in both *PLOD1*-kEDS and *FKBP14*-kEDS (Figure 5G). Bone resorption by osteoclasts is mediated by hypoxic conditions which lead to overexpression of *FGF11* and knockdown of *FGF11* lead to inhibition of bone resorption by osteoclasts in response to hypoxic stimulation [49]. Furthermore, FGF11 protein is strongly expressed in osteoclasts in osteolytic diseases such as rheumatoid synovium and giant cell tumor of bone [49], suggesting that it plays a role in pathological bone resorption. Thus, it will be interesting to determine if FGF11 levels are also elevated in the osteoclasts of kEDS patients, and whether FGF11 is a driver of the bone phenotype in kEDS.

### 4.6. Others

The expression of *PLXNA2* is down-regulated in *PLOD1*-kEDS and *FKBP14*-kEDS patient fibroblasts (Figure 5H). *PLXNA2* encodes the transmembrane protein plexin-A2, a member of the plexin-A family of semaphorin co-receptors, and is demonstrated to be involved in axon guidance [50]. However, nerve conduction appeared normal in *FKBP14*-kEDS patients [3,4] and *PLOD1*-kEDS patients [12,51]. Nevertheless, the suppression of *PLXNA2* expression was of a higher magnitude in *FKBP14*-kEDS patients, in which sensorineural hearing loss has been reported. Hence, whether diminished *PLXNA2* expression has a causal effect on sensorineural hearing impairment in *FKBP14*-kEDS patients requires further investigation. 

In another study, knockdown of *Plxna2* blocked osteoblast differentiation and mineralization in vitro, highlighting the pro-osteogenic role of *Plxna2*. Thus, it may be interesting to investigate whether *PLXNA2* expression is also suppressed in mesenchymal cells of *PLOD1*-kEDS and *FKBP14*-kEDS patients, which may hinder osteoblast differentiation and bone remodeling.

The expression of *PLEKHA2* is diminished in both *PLOD1*-kEDS and *FKBP14*-kEDS patient fibroblasts compared to controls (Figure 5I). *PLEKHA2* expression in dermal fibroblasts of patients with COL6 dominant negative mutations, where mutant type VI collagen chains are incorporated into the collagen fibers, is also significantly lower [10]. *PLEKHA2* encodes pleckstrin homology (PH) domain-containing family A member 2, which is also known as Tandem PH domain-containing protein 2 (TAPP2). TAPP2 is recruited to the plasma membrane via its interaction with phosphatidylinositol-3,4-bisphosphate (PI(3,4)P2) [52,53]. The physiological role of TAPP2 has been well characterized in B-cells, where mice carrying knock-in inactivating mutations of TAPP2 that disrupt its interaction with PI(3,4)P2 develop chronic germinal centers, hyperactive B-cells with higher survival and produce more autoantibodies that worsened with age [54]. Autoimmune phenotypes have, however, not yet been characterized in kEDS patients. Additionally, TAPP2 has also been shown to facilitate the migration of esophageal squamous cell carcinoma cells [55] and malignant B-cells [56] via cytoskeleton reorganization. Hence, further studies on the role of TAPP2 in fibroblast survival, activity, and migration may shed light on its relation to the pathogenesis of kEDS. 

## 5. Conclusions

We have performed the first transcriptomics studies on kEDS which revealed distinct transcriptome signatures between *PLOD1*-kEDS and *FKBP14*-kEDS despite clinical similarities. ECM components are enriched by DEGs in both *PLOD1*-kEDS and *FKBP14*-kEDS, although the genes are unique to each genetic form of kEDS. We aim to validate the candidate DEGs on more patient samples when they become available to strengthen the findings reported here. Proteomics analysis of the ECM component and functional characterization of candidate genes described are currently ongoing and these promise to deepen our understanding of the pathomechanism(s) underlying kEDS.

## Figures and Tables

**Figure 1 genes-10-00517-f001:**
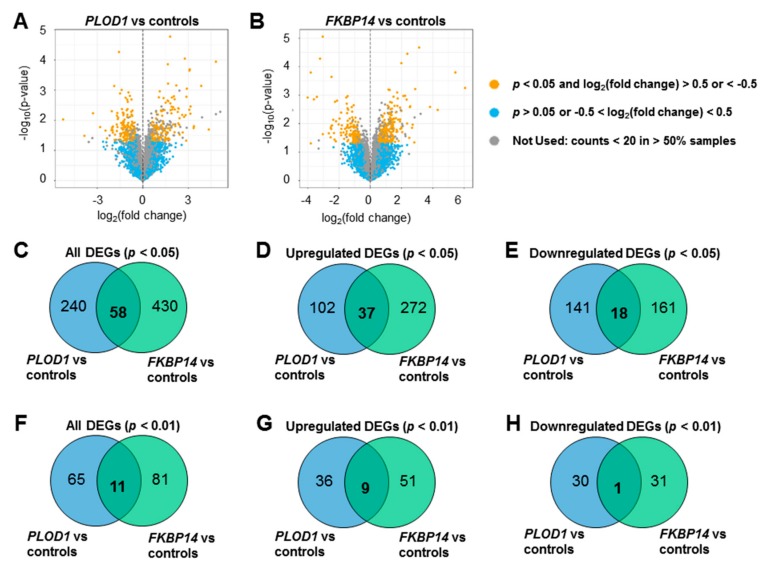
Transcriptome profiles of *PLOD1-kEDS* and *FKBP14-kEDS* patient-derived fibroblasts. Volcano plots of DEGs in (**A**) *PLOD1*-kEDS versus controls and (**B**) *FKBP14*-kEDS versus controls. (**C–E**) Venn diagrams showing number of overlapping and non-overlapping DEGs at *p* < 0.05 (C: all, D: up-regulated, E: down-regulated). (**F**–**H**) Venn diagrams showing number of overlapping and non-overlapping DEGs at *p* < 0.01 (F: all, G: up-regulated, H: down-regulated).

**Figure 2 genes-10-00517-f002:**
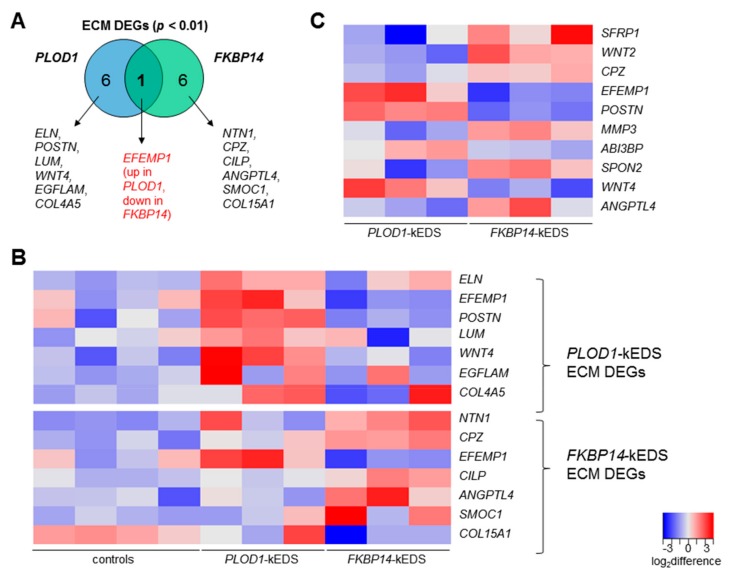
(**A**) Venn diagram depicting unique DEGs contributing to the enrichment of ECM in *PLOD1*-kEDS and *FKBP14*-kEDS patient-derived fibroblasts versus controls. (**B**) Heatmap depicting expression levels of DEGs encoding ECM components per subject. Top panel consists of genes that are differentially regulated in *PLOD1*-kEDS versus controls. Bottom panel consists of genes that are differentially regulated in *FKBP14*-kEDS versus controls. (**C**) Heatmap depicting expression levels of ECM genes that are significantly different between *PLOD1*-kEDS and *FKBP14*-kEDS. The colors indicate the log_2_ difference relative to the average expression of all samples within the comparison.

**Figure 3 genes-10-00517-f003:**
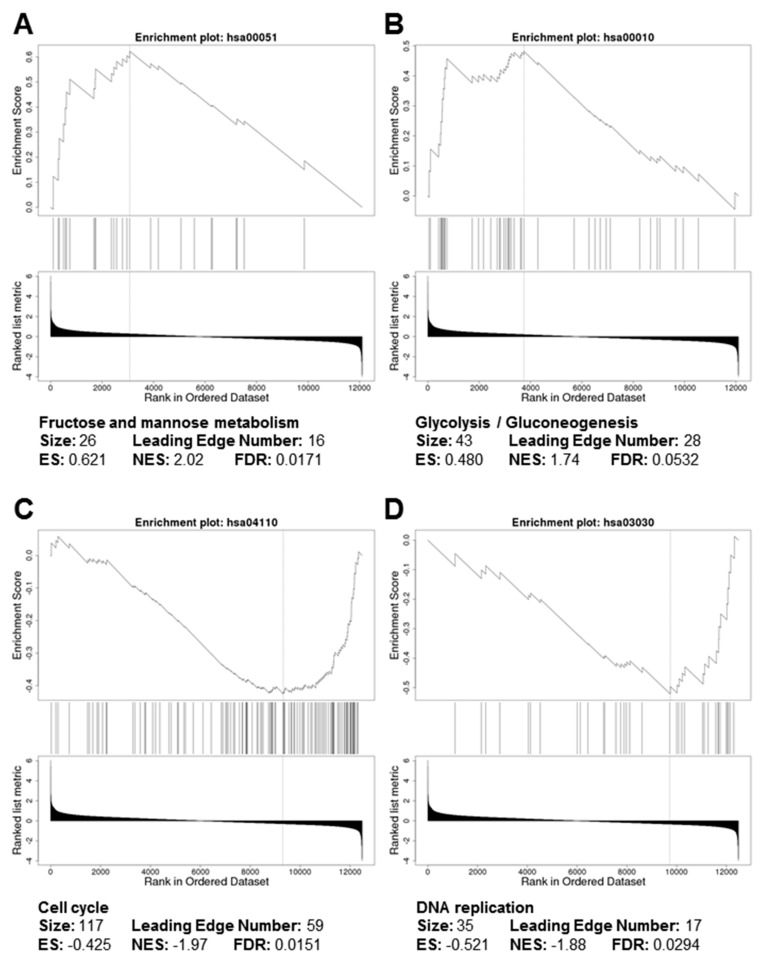
Gene set enrichment analysis (GSEA) plots depicting positive enrichment of (**A**) fructose and mannose metabolism and (**B**) glycolysis / gluconeogenesis and negative enrichment of (**C**) cell cycle and (**D**) DNA replication in *FKBP14*-kEDS versus control fibroblasts.

**Figure 4 genes-10-00517-f004:**
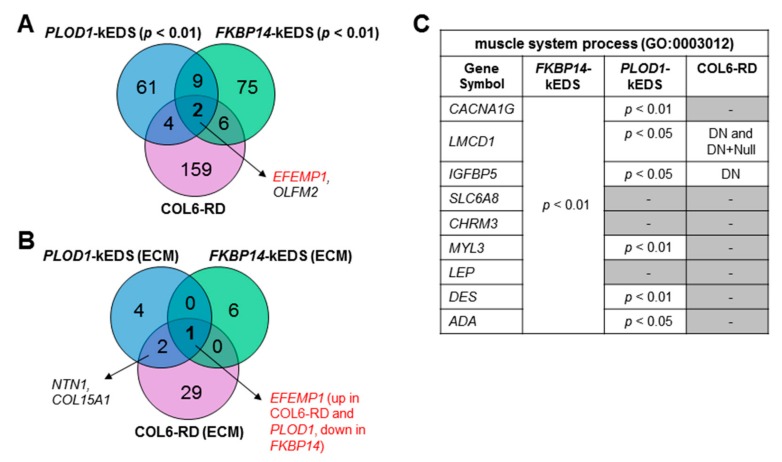
Comparison of transcriptome profiles of *PLOD1*-kEDS, *FKBP14*-kEDS and collagen VI-related muscular dystrophies (COL6-RD) patient-derived fibroblasts. (**A**) Venn diagram depicting number of overlapping and non-overlapping DEGs among three patient groups. (**B**) Venn diagram depicting number of overlapping and non-overlapping DEGs encoding ECM components among three patient groups. (**C**) List of genes contributing to over-representation of muscle system process in *FKBP14*-kEDS versus control fibroblasts and significance of their differential expression in *PLOD1*-kEDS and COL6-RD fibroblasts. DN = dominant negative COL6-RD mutations allowing incorporation of abnormal collagen VI chains; Null = recessive COL6-RD mutations that do not allow incorporation of abnormal chains as described in Butterfield et al., 2017; gene expressions that are non-significantly altered are represented by dashes (-).

**Figure 5 genes-10-00517-f005:**
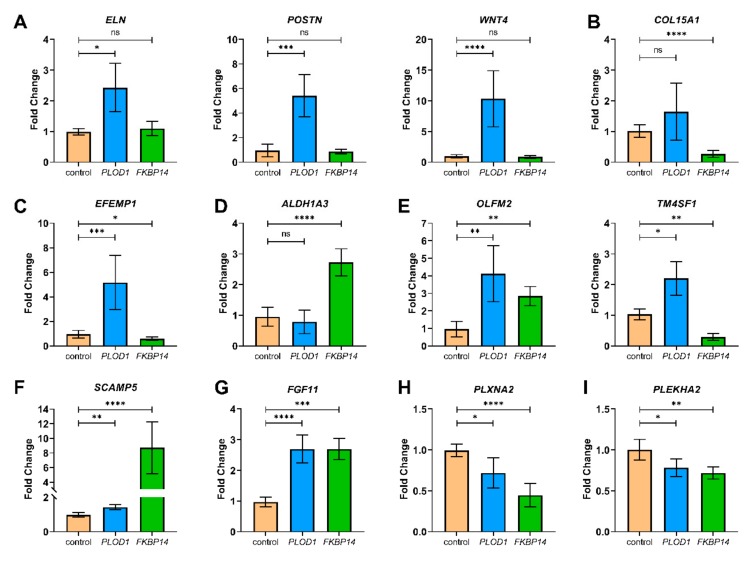
Quantitative RT-PCR was performed to validate the RNA sequencing results. Gene expression levels were measured in four independent replicates per subject, and t-tests were performed (ns = not significant, * = *p* < 0.05, ** *p* < 0.005, *** *p* < 0.0005, **** *p* < 0.0001). Data are expressed as mean ± SEM.

**Figure 6 genes-10-00517-f006:**
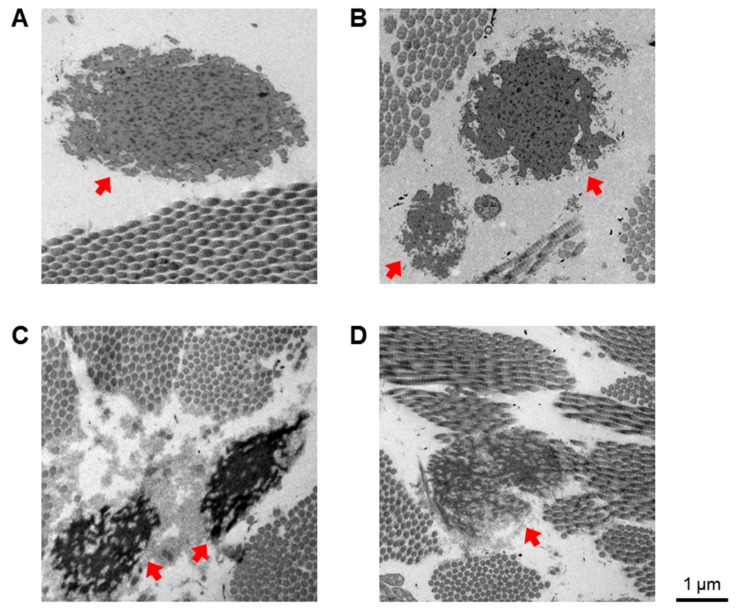
Electron micrographs of skin biopsies from (**A**) a healthy control, (**B**) a *PLOD1*-kEDS patient and (**C**,**D**) two unrelated *FKBP14*-kEDS patients. Elastic fibers are indicated by red arrows. Electron micrographs were taken at the Center for Microscopy and Image Analysis, University of Zurich with a Philips CM 100, a 100 KV transmission electron microscope equipped with a digital charge-coupled device (CCD) camera for image acquisition.

**Table 1 genes-10-00517-t001:** Gene mutations and clinical presentations of *FKBP14*-kEDS and *PLOD1*-kEDS patients included in this study

Gene Mutations	Clinical Findings				
	Hyperextensible Skin	Kyphoscoliosis	Muscle Hypotonia	Joint Hypermobility	Vascular Abnormality
*PLOD1* compound heterozygote c.975+975_1755+?dup/c.1362del p.Glu326_Lys585dup/ p.(Tyr455Thrfs*2) (described as P4 in [11])	+	+ (progressive)	+	+	NR
*PLOD1* p.Glu326_Lys585dup homozygous exon 10-16 duplication (described as P1 in [12])	–	+	+	+	–
*PLOD1* Leu85Pro homozygous (described as P2 in [12])	–	–	–	+	+ (rupture of artery)
*FKBP14* c.362dupC p.(Glu122Argfs*7) homozygous (described as P3 in [4])	+	+ (progressive)	+	+	– (#)
*FKBP14* c.362dupC p.(Glu122Argfs*7) homozygous (described as P1 in [4])	+	+ (progressive)	+	+	–
*FKBP14* c.197 + 5_197 + 8del / p.His67* homozygous (described as P4 in [3])	+	+ (progressive)	+	+	NR

The following symbols and abbreviations are used: +, present; –, absent; NR, not reported. (#) A second-degree cousin died of aortic rupture at age 12, suspected *FKBP14* mutation but DNA not available for confirmation. Another second-degree cousin had a dissection of the internal carotid artery at age 50 years [4]. Please refer to the original publications in which the patients were first described in [3,4,11,12] for more detailed clinical characteristics.

**Table 2 genes-10-00517-t002:** Taqman gene expression assays used for quantitative RT-PCR.

Gene Symbol	Gene Name	Assay ID
*GAPDH*	glyceraldehyde-3-phosphate dehydrogenase	Hs02758991_g1
*ELN*	elastin	Hs00355783_m1
*POSTN*	periostin	Hs01566750_m1
*WNT4*	Wnt family member 4	Hs01573505_m1
*COL15A1*	collagen type XV alpha 1	Hs00266332_m1
*EFEMP1*	EGF-containing Fibulin-like extracellular matrix protein 1	Hs00244575_m1
*ALDH1A3*	aldehyde dehydrogenase 1 family member A3	Hs00167476_m1
*OLFM2*	olfactomedin 2	Hs01017934_m1
*TM4SF1*	transmembrane 4 L six family member 1	Hs01547334_m1
*SCAMP5*	secretory carrier membrane protein 5	Hs01547727_m1
*FGF11*	fibroblast growth factor 11	Hs00182803_m1
*PLXNA2*	plexin A2	Hs00300697_m1
*PLEKHA2*	Pleckstrin homology domain-containing family A member 2	Hs00952489_m1

**Table 3 genes-10-00517-t003:** Over-represented gene ontology terms among DEGs in *FKBP14*-kEDS and *PLOD1*-kEDS patient-derived fibroblasts versus controls.

Gene Ontology	Description	Enrichment Ratio	*p*-value	FDR
***FKBP14*-kEDS: Biological Process**				
GO:0003012	muscle system process	5.52	3.19 × 10^−5^	2.42 × 10^−2^
GO:0007586	digestion	14.1	1.74 × 10^−4^	6.60 × 10^−2^
***FKBP14*-kEDS: Cellular Component**				
GO:0005578	proteinaceous extracellular matrix	5.07	3.90 × 10^−4^	5.77 × 10^−2^
GO:0043235	receptor complex	5.28	8.53 × 10^−4^	6.32 × 10^−2^
***PLOD1*-kEDS: Biological Process**				
GO:0007219	Notch signaling pathway	8.86	1.26 × 10^−5^	9.58 × 10^−3^
GO:0007423	sensory organ development	4.77	3.84 × 10^−5^	1.46 × 10^−2^
GO:0030048	actin filament-based movement	10.7	9.73 × 10^−5^	2.02 × 10^−2^
GO:0045730	respiratory burst	31.2	1.07 × 10^−4^	2.02 × 10^−5^
GO:0060326	cell chemotaxis	7.34	1.56 × 10^−4^	2.38 × 10^−2^
GO:0070098	chemokine-mediated signaling pathway	23.0	2.77 × 10^−4^	3.50 × 10^−2^
GO:0050920	regulation of chemotaxis	6.74	8.43 × 10^−4^	9.16 × 10^−2^
GO:0042490	mechanoreceptor differentiation	14.6	1.10 × 10^−3^	9.61 × 10^−2^
GO:0001655	urogenital system development	4.30	1.14 × 10^−3^	9.61 × 10^−2^
***PLOD1*-kEDS: Cellular Component**				
GO:0005578	proteinaceous extracellular matrix	5.73	1.77 × 10^−4^	2.61 × 10^−2^
GO:1990351	transporter complex	6.36	1.03 × 10^−3^	5.96 × 10^−2^
GO:0044420	extracellular matrix component	7.88	1.56 × 10^−3^	5.96 × 10^−2^
GO:0031594	neuromuscular junction	12.7	1.61 × 10^−3^	5.96 × 10^−2^
GO:0043235	receptor complex	4.98	3.03 × 10^−3^	8.97 × 10^−2^

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
