# Peer review of "Transcriptome Profiling of Primary Skin Fibroblasts Reveal Distinct Molecular Features Between PLOD1- and FKBP14-Kyphoscoliotic Ehlers–Danlos Syndrome"

_genes, 2019, doi:10.3390/genes10070517_

Round 1

Reviewer 1 Report

Pei Jin Lim et al in the manuscript “Transcriptome profiling of primary skin fibroblasts reveal distinct molecular features between PLOD1-andFKBP14-kyphoscoliotic Ehlers-Danlos Syndrome” present the results of a transcriptome analysis of fibroblasts from three patients with PLOD-1 kEDS and three patients with FKBP14-kEDS compared to four healthy controls. The paper presents validation of selected results using RT-PCR. Electron microscopy pictures demonstrating abnormal elastin fibers in FKBP14-kEDS skin is also presented. 

The manuscript is well written. Transcriptome analysis of fibroblasts in kEDS is interesting and important given the rarity of the disease. Yet, the manuscript is descriptive and contains mainly speculations regarding the importance of the results to the phenotype and to the difference between FKBP14-kEDS and PLOD1-kEDS. Conformation at the protein level in the future will be of great interest.

Please see my comments below: 

1) Please add OMIM numbers next to the syndromes and the genes mentioned in the text (at the first time mentioned) 

2) Please avoid using the term 'very rare'. There is no clear quantitative difference between 'rare' and 'very rare'

3) Line 36 is missing a reference to the revised nosology. 

4) Please rephrase the sentence in line 47 so it will be clearer "...increased ratio of the urinary pyridinoline crosslinks hydroxylysyl-pyridinoline and lysyl-pyridinoline(LP/HP) " … lysyl-pyridinoline to hydroxylysyl-pyridinoline…

5) Please add a reference to the process described in the first paragraph of the introduction regarding the role of lysyl hydroxylase in cross linking.

6) Patient P2 is different in the cohort – it is the only patient with homozygous missense variant and missing some of the main characteristics of the syndrome. Interestingly he is the only patient to have a vascular abnormality… Is there a known genotype – phenotype correlation in PLOD-1 related kEDS? The challenge in choosing patents in rare disease research is understood, yet, the reason for this choice of patients should be discussed. 

7) Please add a table (can be as a supplementary table) of detailed clinical data on the patients including demographics (e.g ethnicity, age, gender) and growth parameters. 

8) Please give information regarding the controls used, are they age and sex matched controls?

9) Line 133 -  â€śThe low numbers of DEGs observed in our datasets may be contributed by heterogeneity in multiple factors such as ethnicity, age of biopsy and sex of patients.” To the best of my understanding, when comparing the patients to controls, I would expect that the number of DEGS will be high due to multiple uncontrolled variants such as age, sex and ethnicity and not low. 

10) Line 143 – “more stringent list of DEGs with p<0.01, 11 genes 143 overlapped between PLOD1-kEDS and FKBP14-kEDS fibroblasts (Figure 1F), of which 9 genes were upregulated (Figure 1G) and 1 gene was downregulated (Figure 1H) in both groups”. This sum up to 10 genes. What was the change in the additional gene? 

11) In figure 2c, authors show “Heatmap depicting expression levels of ECM genes that are significantly different between PLOD1-kEDS and FKBP14-kEDS”. It is not clear why this list of genes includes genes that are not present in the list shown in figure 2a (such as WNT 2). To my understanding figure 2a includes all ECM related DEGs in PLOD1-kEDS and FKBP14-kEDS. If a different threshold was used to generate these lists, please clarify.

12) Table 4 should be part of table 3 as it is related to other values seen in table 3.  I believe this will be clearer for the reader.

13) line 198 “collectively highlight different molecular” – given the limitations of this study (including the small sample size), I would suggest changing the general tone to less definitive one and use “suggests that” rather than “highlight”.  The same is true for the statement in line 215. 

14) Please give more information regarding the data published in GEO regarding the specific used dataset. How many patients were tested in the EDS type VI cohort and how many controls? 

15) Quantitative RT-PCR results should be in the results section and not in the discussion. 

16) Figure 5 – are data generated by using 4 technical replicates (from one patient four times)? Is it 4 replicates from each patient (3 biological X 4 technical replicates)? Is the error marked is a SD or SE? Can you please review the calculations of p-values for PLEKHA2? 

17) Line 316 – “These observations highlight the contribution of fibulin-3 to the maintenance of ECM integrity and particularly in the formation of elastic fibers”. This statement has no support in the presented work. The authors present association between the downregulation of fibulin 3 at the RNA level and the presence of aberrations in elastin fibers in the skin of FKBP14-kEDS samples. This does not show any direct effect of fibulin-3 on the elastin fibers or any contribution to the elastin condition. To the best of my understanding, these statements are only speculations at this point. Further experiments at the protein level must be done.

Author Response

 Please see our reply (in red) to your comments below: 

1) Please add OMIM numbers next to the syndromes and the genes mentioned in the text (at the first

time mentioned) 

Revision has been made for this on line 37.

2) Please avoid using the term 'very rare'. There is no clear quantitative difference between 'rare' and 'very rare'

Revision has been made for this on line 37.

3) Line 36 is missing a reference to the revised nosology. 

Revision has been made for this on line 36.

4) Please rephrase the sentence in line 47 so it will be clearer "...increased ratio of the urinary pyridinoline crosslinks hydroxylysyl-pyridinoline and lysyl-pyridinoline(LP/HP) " … lysyl-pyridinoline to hydroxylysyl-pyridinoline…

Revision has been made for this on line 48.

5) Please add a reference to the process described in the first paragraph of the introduction regarding the role of lysyl hydroxylase in cross linking.

Revision has been made for this on line 45.

6) Patient P2 is different in the cohort – it is the only patient with homozygous missense variant and missing some of the main characteristics of the syndrome. Interestingly he is the only patient to have a vascular abnormality… Is there a known genotype – phenotype correlation in PLOD-1 related kEDS? The challenge in choosing patents in rare disease research is understood, yet, the reason for this choice of patients should be discussed. 

No genotype-phenotype correlation in PLOD1-related kEDS has been observed or described. Patient 2 is the only adult patient of the cohort used in this study. Presence of vascular complications has been described only in this patient of the cohort, however it is not an isolated case and there are many example in the literature. The reason why exactly this patient was used is because he is one of the patients diagnosed by us in Zurich and well known to the authors. In our opinion, there is no need to discuss this issue in the paper.

7) Please add a table (can be as a supplementary table) of detailed clinical data on the patients including demographics (e.g ethnicity, age, gender) and growth parameters. 

More extensive clinical data of the patients were reported in the publications (citations 11, 12, 4 and 3) which are now included in Table 1. We have modified the text in Table 1 to clarify this, so that readers can refer to the original publications for more details. The age, sex and origin are now summarised in Table S1.

8) Please give information regarding the controls used, are they age and sex matched controls?

The age, sex and origin of the controls and patients are now summarised in Table S1. Evidently, the age, sex and origin of the patients are variable due to the complexity of the disease severity and progression, making it difficult to find age- and sex-matched controls for the entire study. The average age of our controls are higher than the patients due to limitations in ethical approval and availability of biopsies from healthy children.        

9) Line 133 - â€śThe low numbers of DEGs observed in our datasets may be contributed by heterogeneity in multiple factors such as ethnicity, age of biopsy and sex of patients.” To the best of my understanding, when comparing the patients to controls, I would expect that the number of DEGS will be high due to multiple uncontrolled variants such as age, sex and ethnicity and not low. 

We beg to differ from the reviewer’s opinion. The heterogeneity of the patients contribute to higher deviation within each group of patient such that we obtain a low number of statistically significant DEGs upon setting a p-value threshold.

10) Line 143 – “more stringent list of DEGs with p<0.01, 11 genes 143 overlapped between PLOD1-kEDS and FKBP14-kEDS fibroblasts (Figure 1F), of which 9 genes were upregulated (Figure 1G) and 1 gene was downregulated (Figure 1H) in both groups”. This sum up to 10 genes. What was the change in the additional gene? 

1 of the 11 genes in Figure 1F, EFEMP1, was upregulated in PLOD1 and downregulated in FKBP14. Thus, it does not fall into the category of being upregulated in both PLOD1 and FKBP14 (9 genes in Figure 1G) nor downregulated in both PLOD1 and FKBP14 (1 gene in Figure 1H).

We appreciate that this can be confusing for readers, since both Reviewers 1 and 2 brought this point up. Hence, we have made a comment in the revised manuscript.

11) In figure 2c, authors show “Heatmap depicting expression levels of ECM genes that are significantly different between PLOD1-kEDS and FKBP14-kEDS”. It is not clear why this list of genes includes genes that are not present in the list shown in figure 2a (such as WNT 2). To my understanding figure 2a includes all ECM related DEGs in PLOD1-kEDS and FKBP14-kEDS. If a different threshold was used to generate these lists, please clarify.

Figure 2A depicts differentially expressed ECM genes in PLOD1-kEDS versus controls (7 genes) and in FKBP14-kEDS versus controls (7 genes), of which there was 1 common gene.

In Figure 2C, the comparison was done directly between PLOD1-kEDS and FKBP14-kEDS, and not against controls. Therefore, the genes in 2A and 2C are not the same.

12) Table 4 should be part of table 3 as it is related to other values seen in table 3.  I believe this will be clearer for the reader.

We beg to differ from the reviewer’s opinion. Tables 3 and 4 show different comparisons. Table 3 lists over-represented gene ontology terms among DEGs in FKBP14-kEDS and PLOD1-kEDS patient-derived fibroblasts versus controls while Table 4 lists over-represented gene ontology terms in PLOD1-kEDS versus FKBP14-kEDS.

Considering that both Reviewers 1 and 2 gave the same comment, we decided to remove Table 4 as suggested by reviewer 2  which presents the same idea as Figure 2. Instead, we put the crucial information in Table 4 in the main text on lines 177 to 179 to facilitate the readers.

13) line 198 “collectively highlight different molecular” – given the limitations of this study (including the small sample size), I would suggest changing the general tone to less definitive one and use “suggests that” rather than “highlight”.  The same is true for the statement in line 215. 

Revisions have been made on lines 205 and 222.

14) Please give more information regarding the data published in GEO regarding the specific used dataset. How many patients were tested in the EDS type VI cohort and how many controls? 

Our EDS type VI cohort and controls are described in lines 89-91, and in Tables 1 and S1. We stated the use of 6 EDS type VI patients (3 with PLOD1 mutations and 3 with FKBP14 mutations) and 4 healthy controls. In lines 116 to 119, we mentioned that the RNA-sequencing data, including raw sequence files for each of the 10 subjects will be made available on GEO.

15) Quantitative RT-PCR results should be in the results section and not in the discussion. 

The RT-PCR results have now been moved into the results section as section 3.4.

16) Figure 5 – are data generated by using 4 technical replicates (from one patient four times)? Is it 4 replicates from each patient (3 biological X 4 technical replicates)? Is the error marked is a SD or SE? Can you please review the calculations of p-values for PLEKHA2? 

The data are generated using 4 technical replicates per subject (3 biological X 4 technical replicates for PLOD1-kEDS, 3 biological X 4 technical replicates for FKBP14-kEDS and 4 biological X 4 technical replicates for controls.) The error bars represent standard error of the mean.

We have reviewed the statistics for all data in Figure 5. The previous statistics were done with a one sample t-test for deviation of the mean fold change from 1 for PLOD1-kEDS and for FKBP14-kEDS, which does not take into account deviation within the controls. We have now refined this and performed a t-test to compare the mean fold change between controls and PLOD1-kEDS and between controls and FKBP14-kEDS to better reflect what is shown in the data.

17) Line 316 – “These observations highlight the contribution of fibulin-3 to the maintenance of ECM integrity and particularly in the formation of elastic fibers”. This statement has no support in the presented work. The authors present association between the downregulation of fibulin 3 at the RNA level and the presence of aberrations in elastin fibers in the skin of FKBP14-kEDS samples. This does not show any direct effect of fibulin-3 on the elastin fibers or any contribution to the elastin condition. To the best of my understanding, these statements are only speculations at this point. Further experiments at the protein level must be done.

The reviewer is right that these statements are speculations. We have rephrased our statements in lines 337 – 343 with more caution. We are currently in progress of investigating whether the secretion of fibulin-3 protein by FKBP14-kEDS fibroblasts in vitro is reduced by performing ELISA assays using cell culture supernatant. In addition, since fibroblasts do not form elastin fibers in vitro in 2D cultures, we are also testing if elastin fibers form in an in vitro 3D fibroblast culture system. Should these two points work out well, our next step will be to test if recombinant fibulin-3 proteins can improve elastin fiber formation in vitro.

Reviewer 2 Report

In this study, the authors performed a comprehensive gene expression profiling by transcriptome sequencing (RNA-seq) on cultured skin fibroblasts from patients with recessive mutations in PLOD1 and FKBP14, which are associated with the Kyphoscoliotic Ehlers-Danlos syndrome (kEDS), a rare connective tissue disorder. They recruited 3 kEDS-PLOD1 patients and 3 kEDS-FKBP14 patients, previously characterized for different pathogenic variants in both causative genes. Transcriptome profiling by RNA-sequencing of patients-derived skin fibroblasts revealed differential expression of genes encoding ECM components that are unique between PLOD1-kEDS and FKBP14-kEDS. They also disclosed in both kEDS cell types the differentially expression of genes with functions mainly related to inner ear development, vascular remodeling, ER stress and protein trafficking. qPCR validation of the mRNA levels of a selection of DEGs and electron microscopy analysis of elastic fibers organization from patients’ skin biopsies are also presented.

Given the rarity of the condition, this transcriptome study for the first-time shed light on the pathogenesis of the kEDS. The manuscript is technically sound, well written and contains interesting molecular findings that may help to explain some disease mechanisms related to this rare connective tissue disorder, offering future perspectives for protein validation studies.

I have no major comments; however, there are different minor changes that need to be addressed to improve the quality of manuscript:

· Pag 1 line 36: Please mention the appropriate reference of the corresponding revised nosology.

·   Pag 1 line 37: Please, add the disease OMIM identifiers.

·  Pag 2 line 141: The sentence “We compared DEGs with p<0.05 and log2 (fold change) >0.5 or <-0.5 in both patient groups and found 58 overlapping genes (Figure 1C)” is not clear. In this sentence the total number of common DEGs (n=58) between PLOD1 and FKBP14 deficient fibroblasts, obtained with a p-value<0.05, is different from that showed in the Venn Diagram (Figure 1D, and 1E). Specifically, in the text it is reported that 37 genes are up-regulated (Figure 1D) and 19 down-regulated (Figure 1E), and the sum is 56 and not 58. In addition, in the Venn Diagram the total number of overlapped DEGs, both up-regulated (n=37) and down-regulated (n=18), is 55. Please, correct this mistake (or clarify this point).

· Pag 2 line 143: In the same way, the sentence “Furthermore, from the more stringent list of DEGs with p<0.01, 11 genes overlapped between PLOD1-kEDS and FKBP14-kEDS fibroblasts (Figure 1F), of which 9 genes were upregulated (Figure 1G) and 1 gene was downregulated (Figure 1H) in both groups”, is not clear. Indeed, the total number of common DEGs (by applying a more stringent cut-off, p<0.01) in the text corresponds to 11, whereas in the Venn diagram the number of overlapped DEGs (either up- or down-regulated) is 10 (Figure 1G, 1H). Please correct this mistake (or clarify the discrepancy).

· Pag 3 line 175: Please, specify in the heading of the Table 3 the applied threshold. Moreover, the Table 4 should be part of the Table 3. Alternatively, it could be eliminated, since the corresponding ECM-related genes are illustrated in the Heat Map representation (Figure 2C). This should facilitate the readers.

· Page 7 line 263: A specific paragraph illustrating the qPCR validation of the gene expression changes should be in the result section instead of in the discussion.

Author Response

Please find our reply (in red) to your comments below.

· Pag 1 line 36: Please mention the appropriate reference of the corresponding revised nosology.

Revision has been made for this on line 36.

·   Pag 1 line 37: Please, add the disease OMIM identifiers.

Revision has been made for this on line 37.

·  Pag 2 line 141: The sentence “We compared DEGs with p<0.05 and log2 (fold change) >0.5 or <-0.5 in both patient groups and found 58 overlapping genes (Figure 1C)” is not clear. In this sentence the total number of common DEGs (n=58) between PLOD1 and FKBP14 deficient fibroblasts, obtained with a p-value<0.05, is different from that showed in the Venn Diagram (Figure 1D, and 1E). Specifically, in the text it is reported that 37 genes are up-regulated (Figure 1D) and 19 down-regulated (Figure 1E), and the sum is 56 and not 58. In addition, in the Venn Diagram the total number of overlapped DEGs, both up-regulated (n=37) and down-regulated (n=18), is 55. Please, correct this mistake (or clarify this point).

We thank the reviewer for pointing out an error in the text; Figure 1E is correct. There were 18 genes that were down-regulated in both PLOD1- and FKBP14-kEDS. This is now revised in line 144.

Figure 1C shows that there were 58 common DEGs between PLOD1- and FKBP14-kEDS, regardless of the direction of change. Of these 58 genes, 37 were upregulated in both PLOD1- and FKBP14-kEDS and 18 were downregulated in both PLOD1- and FKBP14-kEDS. The remaining 3 genes were CDA (downregulated in PLOD1, upregulated in FKBP14), EFEMP1 (upregulated in PLOD1, downregulated in FKBP14) and HAND2 (upregulated in PLOD1, downregulated in FKBP14).

· Pag 2 line 143: In the same way, the sentence “Furthermore, from the more stringent list of DEGs with p<0.01, 11 genes overlapped between PLOD1-kEDS and FKBP14-kEDS fibroblasts (Figure 1F), of which 9 genes were upregulated (Figure 1G) and 1 gene was downregulated (Figure 1H) in both groups”, is not clear. Indeed, the total number of common DEGs (by applying a more stringent cut-off, p<0.01) in the text corresponds to 11, whereas in the Venn diagram the number of overlapped DEGs (either up- or down-regulated) is 10 (Figure 1G, 1H). Please correct this mistake (or clarify the discrepancy).

1 of the 11 genes in Figure 1F, EFEMP1, was upregulated in PLOD1 and downregulated in FKBP14. Thus, it does not fall into the category of being upregulated in both PLOD1 and FKBP14 (9 genes in Figure 1G) nor downregulated in both PLOD1 and FKBP14 (1 gene in Figure 1H).

We appreciate that this can be confusing for readers, since both Reviewers 1 and 2 brought this point up. Hence, we have made a comment in the revised manuscript.

· Pag 3 line 175: Please, specify in the heading of the Table 3 the applied threshold. Moreover, the Table 4 should be part of the Table 3. Alternatively, it could be eliminated, since the corresponding ECM-related genes are illustrated in the Heat Map representation (Figure 2C). This should facilitate the readers.

Considering that both Reviewers 1 and 2 gave the same comment, we decided to remove Table 4 as suggested by reviewer 2, which presents the same idea as Figure 2. Instead, we put the crucial information in Table 4 in the main text on lines 177 to 179 to facilitate the readers.

· Page 7 line 263: A specific paragraph illustrating the qPCR validation of the gene expression changes should be in the result section instead of in the discussion.

The RT-PCR results have now been moved into the results section as section 3.4.